# Cell Painting-based bioactivity prediction boosts high-throughput screening hit-rates and compound diversity

Johan Fredin Haslum[1,2,3], Charles-Hugues Lardeau[4], Johan Karlsson[5], Riku Turkki[5], Karl-Johan Leuchowius [5], Kevin Smith[1,2] & Erik Müllers [3] ✉

Identifying active compounds for a target is a time- and resource-intensive task in early drug discovery. Accurate bioactivity prediction using morphological profiles could streamline the process, enabling smaller, more focused compound screens. We investigate the potential of deep learning on unrefined single-concentration activity readouts and Cell Painting data, to predict compound activity across 140 diverse assays. We observe an average ROC-AUC of 0.744 ± 0.108 with 62% of assays achieving ≥0.7, 30% ≥0.8, and 7% ≥0.9. In many cases, the high prediction performance can be achieved using only brightfield images instead of multichannel fluorescence images. A comprehensive analysis shows that Cell Painting-based bioactivity prediction is robust across assay types, technologies, and target classes, with cell-based assays and kinase targets being particularly well-suited for prediction. Experimental validation confirms the enrichment of active compounds. Our findings indicate that models trained on Cell Painting data, combined with a small set of single-concentration data points, can reliably predict the activity of a compound library across diverse targets and assays while maintaining high hit rates and scaffold diversity. This approach has the potential to reduce the size of screening campaigns, saving time and resources, and enabling primary screening with more complex assays.

Drug discovery campaigns typically rely on high throughput screening (HTS) for hit finding i.e., the process of identifying and selecting chemical compounds with biological activity towards a target and the potential to be developed into a drug. Such screens can involve probing millions of compounds. Although the throughput of these types of experiments has significantly increased thanks to technological advancements in automation and robotics, it is still a time and resource intensive process. Because of this, hit finding is generally done with simple assays such as biochemical assays to enrich the compound set before more resource-intense assays can be used further down the cascade. Because the initial screening assays are often

very simple representations of the target biology, they run the risk of producing false positive and negative results. Whereas false positives can be identified and removed by further probing with follow-up assays, false negatives can be problematic as they can filter out potentially interesting compounds. Thus, there is an interest in using as biologically relevant assays as possible early in the screening cascade.

One strategy to accelerate hit finding is to use computational methods to prioritize and select compounds deemed more likely to be active. Predicting bioactivity has been used to enrich compound sets in HTS assays[1,2]. These types of approaches promise to efficiently enrich

[1]KTH Royal Institute of Technology, Stockholm, Sweden. [2]Science for Life Laboratory, Stockholm, Sweden. [3]Research and Early Development, Cardiovascular, Renal and Metabolism (CVRM), BioPharmaceuticals R&D, AstraZeneca, Gothenburg, Sweden. [4]Discovery Sciences, R&D, AstraZeneca, Alderley Park, UK. [5]Discovery Sciences, R&D, AstraZeneca, Gothenburg, Sweden. ✉e-mail: mullers.erik@gmail.com

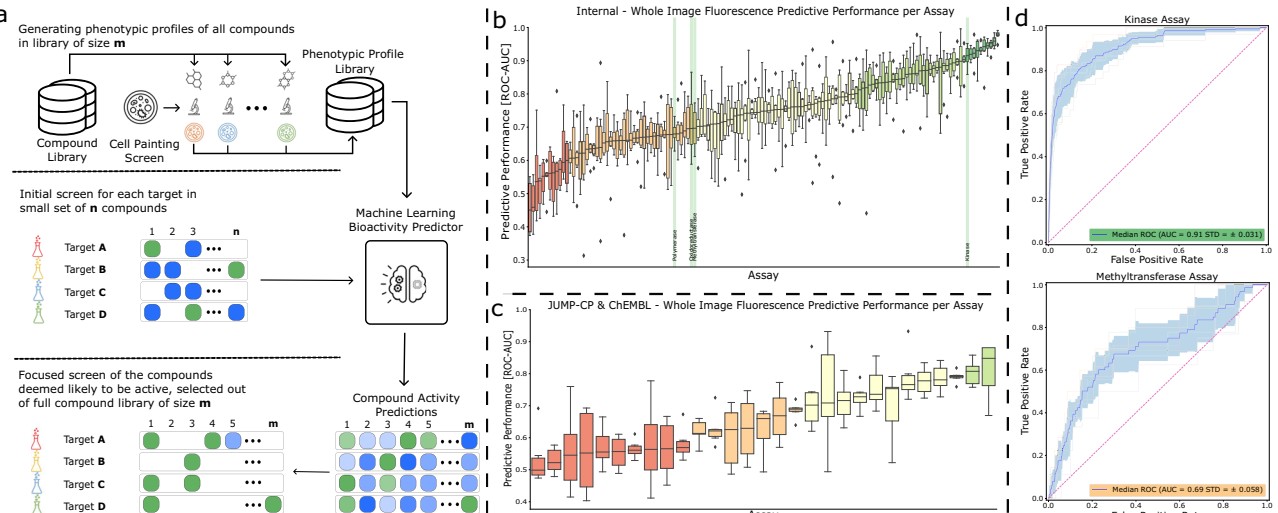

**Fig. 1 | Using phenotypic screening for bioactivity prediction. a** The envisioned approach utilizing phenotypic screening for bioactivity prediction. A single High Content Imaging screen using Cell Painting is used to generate phenotypic representations of each compound in the compound library m - $10^6$. Small scale screens, $n$ - $10^4$ for a target of interest are then used to generate activity readouts for a set of compounds used to train a machine learning model (green active, blue inactive). This model is then used to rank compounds for probability of activity and prioritize compounds for further screening (green predicted active, blue predicted inactive). **b** Boxplot of ROC-AUC performances of each assay in each cross-validation test-split ($n = 6$ splits), ranked in order of median ROC-AUC performance, center value defining median with boxes indicating the IQR and whiskers extending to extreme points or maximally ±1.5 × IQR. Highlighted assays in green are investigated in secondary screens. **c** Boxplot of ROC-AUC performance of each assay in each cross-validation test-split ($n = 6$ splits) for the JUMP-CP data, ranked in order of median ROC-AUC performance, center value defining median with boxes indicating the IQR and whiskers extending to extreme points or maximally ±1.5 × IQR. **d** Receiver Operating Characteristic curve of two example assays. The dark blue line represents the average ROC-curve, the shaded area represents the standard-deviation intervals and the faded lines ROC-curves of individual cross-validation splits. Source data are provided as a Source Data file.

likely hit compounds into focused compound sets. Because these focussed compound sets contain much fewer compounds than the full compound libraries, they could enable earlier use of assays with higher biological relevance, e.g., iPSC-derived, or primary cells, which are typically restricted to later stage drug discovery due to cost and/or scarcity of biological material.

So-called structure activity relation (SAR) models are a family of computational methods, used to make bioactivity predictions or property predictions i.e., using computational methods and models to estimate bioactivity or properties of chemical compounds. These models rely on compound structure information to make predictions of compound activity on a particular target. Recently, alternatives to structure representations have been explored[2–5] for prediction of bioactivity[6] or toxicology[7]. One such example is the use of phenotypic profiles[6]. Phenotypic profiles are derived from cells, tissues, or even whole organisms, and contain information on the characteristics or behaviors of these complex biological systems in response to perturbations with small molecule compounds or other drug modalities. Phenotypic profiles can encompass a wide range of biological responses, including changes in cell morphology, proliferation, gene or protein expression, and physiological functions. Compound bioactivity prediction and compound property prediction by phenotypic profiling have emerged as attractive alternatives to SAR as they have proven to be capable of enriching compound sets while at the same time alleviating some of the drawbacks of structure-based models, such as low structural diversity and limited scaffold-hopping potential.

While it has been shown that information present in phenotypic screens can be used to predict bioactivity in unrelated targets[6,8], previous approaches have relied on dose-response data and/or multiple data points per compound as activity readouts. However, in practice, such data are usually not available at the scale required at the early hit identification stage – where the machine learning approaches proposed in these works can be meaningful applied.

In this work, we adopt a more practical setting, that better reflects the reality of early hit finding, utilizing unrefined single-concentration activity readouts which are relatively cheap and easy to produce and are often readily available at the early hit identification stage. In combination with these data, we rely on Cell Painting images of the full set of compounds – an additional cost – but one which only needs to be produced once and can then be reused across all the different assays and targets we want to predict (Fig. 1a). We employ a large-scale general purpose Cell Painting screen to capture phenotypic profiles of a library of available compounds and train a model using small, focused bioactivity assay readouts for specific targets. The trained model is then used to predict the bioactivity of the compounds in the entire compound library, enabling the selection of compounds most likely to modulate the intended target (Fig. 1a). This approach has the potential to reduce the number of compounds to be screened, as well as the number of assays and experiments required in early drug screening cascades. As only a few hundred activity data points are needed to train the predictive model for a particular target and assay, assays of higher complexity and biological relevance could potentially be used. Furthermore, we explore different input modalities for bioactivity prediction, including fluorescence images, brightfield images, and image features extracted from the fluorescence images using classical image analysis approaches. Comparing these image-based approaches to traditional structure-based approaches, we demonstrate that Cell Painting-based compound bioactivity prediction can outperform structure-based approaches in their predictive performance as well as the structural diversity of the top ranked compounds. To further strengthen the practicality of our approach, we analyze prediction performance and robustness across various assay types, technologies, and target classes to identify specific targets and assays that are particularly well-suited for bioactivity prediction. Finally, we confirm the validity of our predictions through a series of in vitro follow-up experiments which demonstrate that the bioactivity predictions of our models are reliable and consistent.

Our results demonstrate the capability of models trained on phenotypic data combined with a few hundred single-concentration data points, to predict compound activity reliably and efficiently across diverse targets in a realistic drug screening scenario.

## Results

### Phenotypic bioactivity prediction across 140 diverse assays

We selected a structurally diverse set of 8,300 compounds to be representative of a larger HTS screening library. This subset was screened in a Cell Painting assay, an optimized high-content microscopy assay that utilizes a set of six fluorescent dyes to label different cellular components, including the nucleus, nucleoli, endoplasmic reticulum, mitochondria, cytoskeleton, Golgi apparatus, plasma membrane, actin filaments, as well as cytoplasmic and nucleolar RNA[9]. For each compound we also extracted corresponding single-point bioactivity data from the AstraZeneca HTS database. In this context, each activity label is based on a single data point, representing a unique compound at a specific concentration in a single microwell from an HTS screen. The resulting dataset consisted of 8,300 compounds with Cell Painting images of compound-treated cells and associated single concentration bioactivity data in one or more of 140 unique assays, see Supplementary Fig. 1 for more assay details. Each of those assays had at least 50 active compounds. The label matrix, according to compounds and assays, had a 47.8% fill rate and an average of 3% of the known compounds labeled as active (for more details of the data selection see Materials and Methods).

We split the data into six folds, with each compound appearing only in a single fold. Compounds that were structurally similar, based on ECFP-4 clustering, were assigned to the same fold to measure the ability of the model to identify actives in unknown regions of the compound space. This practice is commonly performed in Structure Activity Relationship (SAR) modeling. We applied nested cross-validation to the data, training a ResNet50[10] in a supervised multi-task learning setup. The model was pretrained using ImageNet[11] and modified to accept 5-channel fluorescence images as input. It was then trained to predict bioactivity readouts for each of the 140 assays. In the cross-validation process, four of the data folds were used to train the model, one fold was used as validation to tune hyper-parameters, and one fold was used as a test set to evaluate performance (see Materials and Methods for details on training and hyperparameter selection).

The performance of the model, averaged across the six folds, was measured at a ROC-AUC of $0.744 \pm 0.108$. Performance varied between the assays (Fig. 1b) and we found a slight correlation between performance and number of known compounds and actives (Supplementary Fig. 2). Whereas predictions for many assays showed low correlation, predictions for some assays were correlated or anti-correlated (Supplementary Fig. 3), which appeared unrelated to the predictive performance, assay type, technology, or target type. 62% of the assays achieved an ROC-AUC of 0.7 or more, which we deem good performance, while 30% reached 0.8 or higher (indicating very good performance), and a further 7% reached 0.9 or higher (indicating excellent performance). Overall, these results indicated that Cell Painting data contains valuable information related to bioactivity for a wide range of target and assay types. This relationship can be learned by a deep learning model using small sets of single-concentration activity readouts.

To validate the reproducibility of our approach, we applied the same predictive framework to two publicly available datasets. Initially, we assessed our framework's performance on a dataset established by Hofmarcher and colleagues[8], demonstrating end-to-end learning with convolutional neural networks (CNNs) for biological assay prediction from Cell painting images. This dataset included 209 assays comprising 10,574 compounds[8,12], where binary activity data was derived from dose-response curves (IC50/EC50) of each compound in a given assay. Employing identical cross-validation splits, our model exhibited an average performance of $0.731 \pm 0.198$ ROC-AUC across these 209 tasks, with 116 assays achieving ≥0.7, 84 reaching ≥0.8, and 68 surpassing ≥0.9 (Supplementary Fig. 4). Notably, our results align closely with the performance reported for the supervised ResNet model by Hofmarcher et al. ($0.731 \pm 0.19$ ROC-AUC)[8] and the linear probing contrastive learning model (CLOOME) recently reported on the same dataset ($0.714 \pm 0.20$ ROC-AUC)[12]. Subsequently, we created a dataset employing a subset of the recently published JUMP-CP dataset[13], in combination with activity data from ChEMBL[14]. This subset encompassed 29 assays comprising 10,660 unique compounds (See Materials and Methods 1.3. JUMP consortium and ChEMBL datasets for details). The average performance for these 29 assays was $0.660 \pm 0.094$ ROC-AUC. Detailed performance for each of the 29 assays can be found in Fig. 1c.

### Comparing bioactivity predictions using different input modalities

As described above, we observed encouraging results using a multiplexed fluorescence Cell Painting screen to capture phenotypic profiles of a library of compounds. As it has been shown that brightfield images can be used to predict Cell Painting features[15], we wanted to investigate if the information content in the images would be sufficient to predict bioactivity of compounds. Brightfield imaging has some advantages compared to Cell Painting-stained cells as it can be performed on live cells and does not require staining of the cells and can be performed on simpler microscopes. These factors could significantly reduce the cost of the assay and enable kinetic assays, although potentially at the expense of less informative image data.

We also wanted to investigate whether the end-to-end features learned by the neural network in our setup had an advantage over hand-crafted features. To this end, we extracted hand-crafted image features, hereafter referred to as Cell-Features, using the Columbus image-analysis software. Similar features could be extracted using free software such as CellProfiler[16]. We compared the performance of the neural network trained with Cell Painting images (labeled Whole Image Fluorescence) versus a similar network trained with brightfield images (Whole Image Brightfield), as well as a neural network trained on hand-crafted cell features (labeled Cell-Features). These image-based modalities were then compared against a standard structure-based approach using Extended Connectivity Fingerprints[17] (labeled Structure). Each model was assessed using the same cross-validation splits as described above. The fluorescence and brightfield images were used to train ResNet50 models, while the cell-features and structure-based data were used to train multi-layer perceptrons (See Materials and Methods for details).

Evaluation on the held-out test sets revealed that the predictive performance of the whole image fluorescence-based approach outperformed all other approaches (Fig. 2a). This approach reached an average ROC-AUC of $0.744 \pm 0.108$ compared to the cell-feature based model at $0.726 \pm 0.115$. Both fluorescence-based approaches (Whole Image Fluorescence and Cell-Features) outperformed the brightfield model at $0.704 \pm 0.107$ ROC-AUC. The structure-based approach performed the worst at $0.686 \pm 0.100$. Statistical analysis using Friedman rank sum test with the assay as blocking factor revealed significant performance differences ($p = 4.47 \times 10^{-19}$) between the modalities (Fig. 2a). Applying a post-hoc Nemenyi's test, we find that the performance differences are significant between all modalities except for brightfield and structure. Although the brightfield image-based approach was outperformed by the fluorescence-based approach, it was still able to predict 49% of the assays with a ROC-AUC above 0.7 and even 5% above 0.9. This shows that the information captured in brightfield images can be linked to bioactivity in a wide range of assays and targets, which may justify using brightfield images in some cases despite their slightly inferior performance. We also tried to combine the brightfield images with the fluorescence images to see if it would

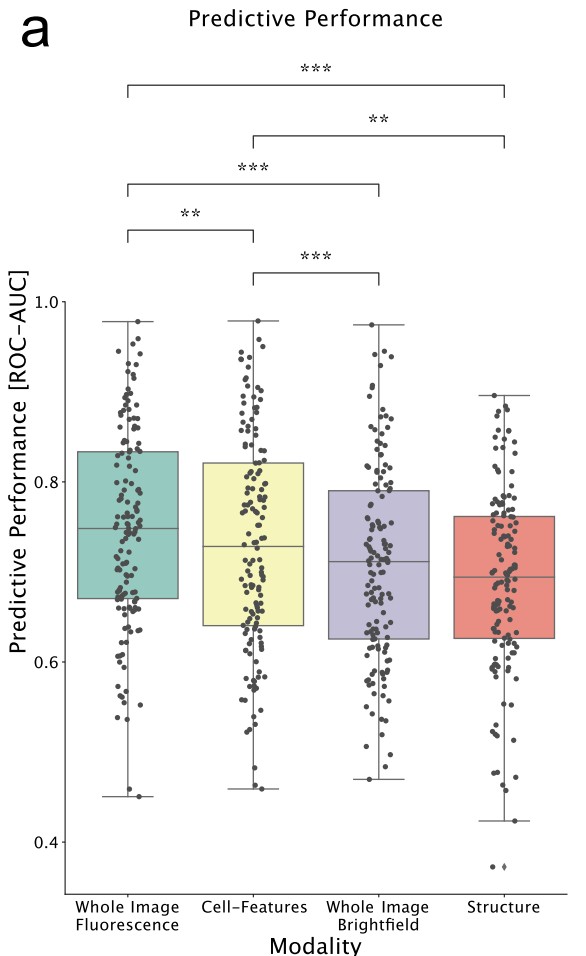

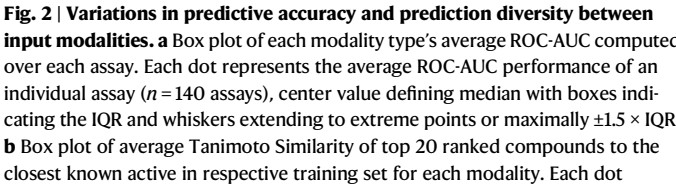

**Fig. 2 | Variations in predictive accuracy and prediction diversity between input modalities. a** Box plot of each modality type's average ROC-AUC computed over each assay. Each dot represents the average ROC-AUC performance of an individual assay ($n = 140$ assays), center value defining median with boxes indicating the IQR and whiskers extending to extreme points or maximally $\pm1.5 \times$ IQR. **b** Box plot of average Tanimoto Similarity of top 20 ranked compounds to the closest known active in respective training set for each modality. Each dot represents the average Tanimoto similarity score per assay over all cross-validation splits. Only assays with average performance above 0.6 ROC-AUC in all modalities were included ($n = 87$ assays), center value defining median with boxes indicating the IQR and whiskers extending to extreme points or maximally $\pm1.5 \times$ IQR. Statistical analysis was done using Nemenyi's-Friedman post-hoc test, two-sided, ** representing $10^{-3} < p < 10^{-2}$, *** representing $10^{-4} < p < 10^{-3}$. Source data are provided as a Source Data file.

offer any improvements; however, we saw no significant improvement in the predictive performance, indicating that the brightfield images contained little complementary information to the fluorescence images.

A natural question to ask when predicting bioactivity of a compound library is: how chemically diverse are the top predictions? For each bioactivity prediction approach, we compared the structural diversity of the 20 top-ranked compounds to the known actives in the training set. Our analysis revealed that compounds predicted from images showed lower structural similarity i.e., greater chemical diversity, than structure-based approaches. A Friedman rank sum test reveals a significant difference in the distribution of compound diversities ($p = 4.21 \times 10^{-12}$). A Nemenyi's post-hoc test showed that the chemical diversity of structure-based predictions was significantly lower than image-based predictions (Fig. 2b).

Overall, the Cell Painting fluorescence-based approach performed best both in terms of correctly predicting bioactivity (measured by ROC-AUC) and in terms of increased chemical diversity (measured by Tanimoto similarity). The Cell-Features approach, using extracted image features, resulted in a slightly lower performance than the Whole-Image Fluorescence approach. The Whole-Image Brightfield approach also resulted in a slightly lower performance. However, it

may still be an attractive alternative because the slight drop in predictive performance can be justified by other benefits compared to the Cell Painting assay.

**The impact of assay and target characteristics on performance**

As seen in Fig. 1b, the predictive performance of the Cell Painting fluorescence-image based model varied widely from assay to assay, ranging from 0.96 to 0.48 ROC-AUC. We wanted to understand the factors influencing this performance variation and to identify which conditions are particularly suited for bioactivity prediction. Therefore, we conducted a detailed analysis, breaking down the results to examine how various assay characteristics contribute to performance (Fig. 3).

The general trend we found was that the predictive performance was consistently good for different assay types. Likewise, predictive performance was consistently good across target types, therapy areas, and assay technologies. Nevertheless, some noteworthy insights emerged: the fluorescence-based methods were better at predicting cell-based assays than biochemical assays, and the structure-based method was better at predicting biochemical assays than cell-based assays (Fig. 3a). Among molecular target subtypes, kinase targets appeared to benefit the most from our Cell Painting-based approach,

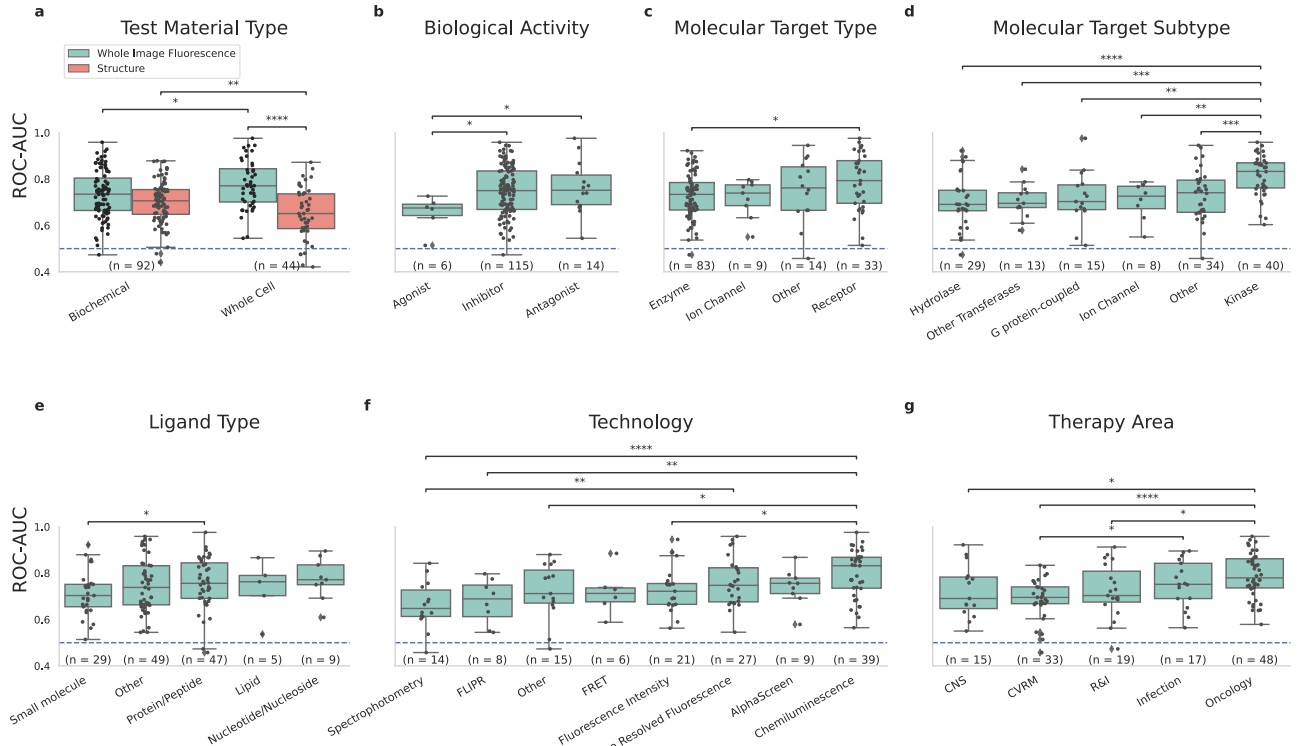

**Fig. 3 | Predictive performance variations across assay characteristics.** Predictive performance of various assays carried out in our experiments, grouped by assay characteristics. Shown as box plots, center value defining median with boxes indicating the IQR and whiskers extending to extreme points or maximally ±1.5 × IQR. **a** The predictive performance of the image-based Fluorescence model compared to the structure based, when grouped by Test Material Type. Comparison of Assay performance of the image-based Fluorescence model grouped by assay characteristics: **b** Biological Activity, **c** Molecular Target Type, **d** Molecular Target Subtype, **e** Ligand Type, **f** Assay Technology and **g** Therapy Area. R&I Respiratory & Immunology, CVRM Cardiovascular renal metabolism. Significance values calculated using Kruskal-Wallis test, followed by Conover pair-wise test, * representing $10^{-2} < p < 5*10^{-2}$, ** representing $10^{-3} < p < 10^{-2}$, *** representing $10^{-4} < p < 10^{-3}$, **** representing $p < 10^{-4}$. Source data are provided as a Source Data file.

performing significantly better than other molecular target subtypes (Fig. 3d). The increased performance on kinase targets could be attributed to the known promiscuity of kinase inhibitors, which can affect multiple cellular pathways, leading to stronger phenotypic responses that are more readily identified by the model. In terms of assay technologies (Fig. 3f), predictions were especially accurate in chemiluminescence assays, and less so in assays using spectrophotometry. Among the therapy areas covered in our experiments, performance in oncology assays was significantly better than other areas, possibly due to a higher fraction of kinase targets in that therapy area.

## Follow-up assays validate fluorescence-based predictions

Previous studies that have aimed to predict bioactivity from image-based assays have limited their analysis to a single primary assay. But biological and technical noise in the process can lead to inaccurate and potentially overoptimistic results when the experiment is repeated. We put the predictions of our Cell Painting-based model to the test by running secondary assays for the same targets. We selected follow-up assays that were used in the screening cascades as secondary assays to triage HTS hits, spanning different assay technologies and different target classes: a methyltransferase, a polymerase, an oxidoreductase, and a serine kinase. The follow-up assays were chosen to represent a range of performances in the primary assay, from a low ROC-AUC of $0.68 \pm 0.032$ up to a very high performance of $0.91 \pm 0.031$. The assays selected for follow-up are marked in Fig. 1b. In each of the selected follow-up assays, a ranked list of predicted bioactivities was produced by our model. The majority of the top-ranked 5% of compounds were randomly sampled and included in the follow-up assay, along with

selection of compounds selected uniformly at random (at least 1000 compounds in total). The ROC-AUC reported in our experiments was computed using only the randomly selected compounds to keep the values comparable with the primary assay. The top-ranked compounds were included to make meaningful measurements of enrichment as it was expected that few of the randomly selected compounds would be active, due to the low assay hit rates (i.e., out of 500 randomly selected compounds, only 5-10 were expected to be active).

We found the ROC-AUC values in the follow-up assays to be consistent with the values from the primary assays. In fact, two out of the four follow-up assays performed slightly better than their respective primary assays (Fig. 4a and Supplementary Table 1). Our results suggested that not only do the model predictions carry over to follow-up experiments, but that the expected range of performance is consistent as well.

While ROC-AUC helps understand the predictive performance of the model, ultimately, what we care about is how the model can be used to enrich the compound sets. To calculate enrichment, using bootstrapping we probed the top 5% of ranked compounds for each assay (accordingly, the theoretical maximum enrichment from this experiment is 20x). Overall, the follow-up assays showed enrichment values in line with their predictive performance, often better (Fig. 4b and Supplementary Table 1). The serine kinase assay, which had the highest predictive performance (ROC-AUC 0.91) showed an astoundingly high enrichment of 14x in the follow-up, representing a significant improvement and suggests this assay could focus on a small, highly targeted set of compounds. The other assays, Oxidoreductase, Polymerase, and Methyltransferase, all had follow-up enrichments in line with the primary assay (6.4x, 4.6x, and 1.6x respectively). The

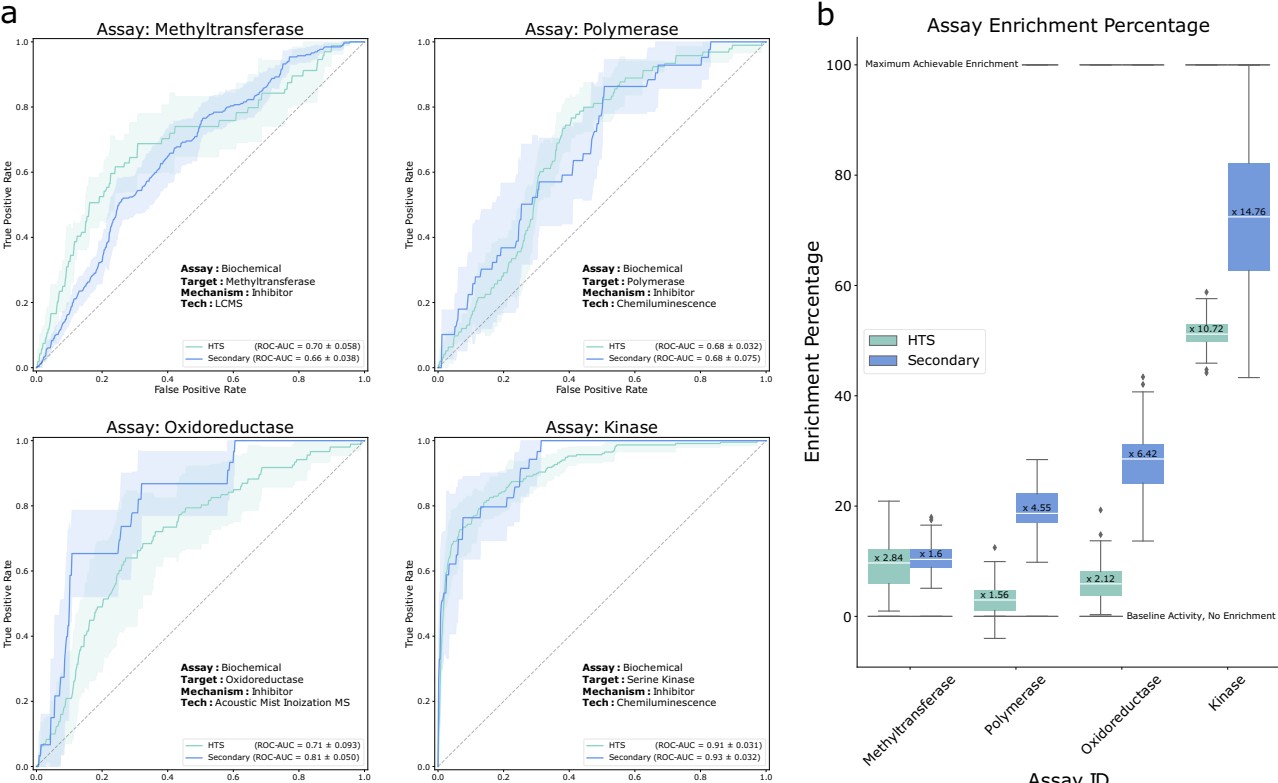

**Fig. 4 | Validating model prediction in secondary screen.** Model validation using follow-up screens in four of the assays. Top ranked compounds suggested from the fluorescence image-based models were screened in corresponding secondary assays (see text for full description). **a** Receiver operating curves for the four assays, primary HTS (green) and follow-up (blue), shaded area represents standard-deviation interval. ROC-AUC values for each of the four assays were calculated using the randomly sampled subset of compounds, green showing the average performance in the original HTS assays and blue representing the performance when using secondary screen activity readouts. **b** Enrichment values of the top 5% predicted compounds for each of the four assays. Enrichment was calculated for the HTS (green) and secondary (blue) activity readouts. Box plots ($n = 100$ boot-strapped), center value defining median with boxes indicating the IQR and whiskers extending to extreme points or maximally ±1.5 × IQR. Source data are provided as a Source Data file.

follow-ups for Oxidoreductase and Polymerase showed higher enrichment than the primary assay, while the follow-up for Methyltransferase showed a slightly lower enrichment. However, it should be noted that the ROC-AUC was consistent between the primary and secondary. The weaker enrichment may be explained by the high baseline bioactivity in the Methyltransferase follow up assay, which limited the theoretical maximum enrichment of this assay to approximately 6×.

Several of the top ranked compounds were shown to be highly potent (low nM potency), indicating the quality of our predicted hits was high. Moreover, several compounds were predicted to be active by the model and were confirmed to be active, despite them having been labeled as inactive in the original HTS data. Their activity could be further confirmed in orthogonal assays, highlighting the robustness of our predictions, and indicating that our model offers opportunities to rescue false-negative compounds. In summary, the assays probed in our follow-up experiments showed that the model performance was conserved through replication, even perhaps slightly better than what was expected. This indicates that the model's predictions are driven by the targets and phenotypes, and not significantly affected by biases and noise in the assays. Furthermore, the enrichment levels we observed were high enough to reduce cost and speed up the screening process by filtering in silico compounds according to the ones the model predicts to be active.

## Discussion

Hit finding in early drug-discovery is time and resource intensive, often relying on target-specific HTS campaigns to identify diverse, bioactive compounds. We assessed the possibility of rationalizing the hit finding process using phenotypic-based bioactivity prediction models, trained with unrefined single-point activity data. These types of data are relatively inexpensive to produce and can therefore readily be used as a basis for training the prediction models for targets. This facilitates the practical use of these types of models in early drug discovery pipelines, enabling the creation of focussed compound sets containing compounds predicted to be bioactive.

Our results show that a model trained using phenotypic data from a single general-purpose Cell Painting screen can predict bioactivity in a wide range of assays, outperforming commonly used SAR models in terms of both predictive performance and structure diversity. We validated the model's performance in follow-up experiments in secondary assays. The results showed that Cell Painting-based bioactivity prediction using morphological profiles was feasible for a wide range of targets. Our approach has the potential to reduce the number of compounds screened, as well as the number of assays and experiments required in drug screening cascades, which in turn could allow for early screening of focused compound sets in assays of higher biological relevance. In addition, our results show that even a brightfield-based model can perform slightly better than structure-based predictions, while also identifying more diverse compounds. This is in line with previous work showing that much of the information content of fluorescence images can be inferred from brightfield images[15,18,19] and recent work by Harrison et al. that found largely correlated predictive performance for fluorescence and brightfield[20]. While the predictive performance of brightfield images does not fully reach the level of fluorescence images, brightfield offers a cheaper alternative for

phenotypic profiling. Moreover, the use of brightfield images also opens the possibility of live cell imaging to include temporal information of compound effects in cells.

Previous work by Simm et al. and Hofmarcher et al. established the information link between phenotypic screening data and assay activity[6,8]. These early studies employed binary activity data derived from dose-response curves, expressed as pXC50[6] or IC50/EC50[8] of the given compound in the given assay. In these cases, each activity label is often supported by multiple data points (usually 10–20), significantly enhancing the confidence in the labels. In addition, the previous approaches evaluated assays at 3–4 thresholds of pXC50/IC50/EC50. Consequently, a relatively low level of noise in the labels is expected. Our results extend this by demonstrating the capacity to learn from small sets of readily available, but relatively noisy, unrefined single-point activity readouts. Here each binary activity label is based on a single data point from a single concentration of a compound in a single well of a bioassay, as is common in high-throughput screens. We demonstrate efficient learning despite the inherent noise in single-point activity data. Our approach could significantly reduce the size of screening campaigns, saving time and resources, and enabling primary screening with more complex assays or scarce material.

While structure-based bioactivity prediction is attractive as it requires no in vitro data, alternative input representation can avoid the problems SAR models have with scaffold hopping and increasing the diversity amongst predicted hits. Previous work using activity[4,5], transcriptomics[21] and phenotypic fingerprints[6,8] are examples of such approaches.

There has been limited work comparing models based on chemical structure and those based on imaging data[22]. Concurrent work has shown that cell feature-based model performance is often comparable[23] or slightly superior[24], aligning with our observations. Several factors appear to influence these models' comparative performance such as the compound set and associated data splitting strategy, and the specific assay characteristics. Our image-based model on average outperforms a structure-based model. Nonetheless, noticeable performance variations exist among different assay and target types, as illustrated in Fig. 3a. The performance difference between structure- and image-based model is significant for cell-based assays but does not reach significance for biochemical assays. Recent advancements in structure-based modeling, including techniques like Graph Neural Networks (GNNs)[25] and Transformer-based models[26], hold the potential for further enhancing performance.

Recently there has been a strong interest in combining compound structure information with activity fingerprints leading to improved performance in bioactivity prediction[4]. The same was shown to be true for cell state, where gene-expression and phenotypic data provided complementary information boosting performance prediction[27]. More recent work has also explored the use of phenotypic data combined with other modalities in supervised or contrastive learning settings for better bioactivity prediction[24,28], toxicity prediction[29], and Mechanism of Action prediction[30,31]. Beyond the inclusion of auxiliary input modalities, performance can likely be improved by employing more powerful deep learning model types[32,33], for example, state-of-the-art deep learning networks such as Vision-Transformers, which have showed improved performance in medical imaging tasks[34]. Other training strategies such as Self-Supervised Learning pretraining could also have the potential to further improve performance.

To validate our predictions and understand the level of enrichment and its relation to the primary assay, we selected four diverse assays with predictive performance ranging from a low ROC-AUC of 0.68 to a very high performance of 0.91. We sampled and tested at least 1,000 compounds for each assay, encompassing the majority of the top-ranked 5% of compounds, along with at least 500 compounds selected uniformly at random. Employing bootstrapping, we consistently observed enrichment ranging from a moderate 1.6-fold to a high 14-fold, correlating with the predictive performance of the assays. In a prior study by Simm et al., 342 top-ranked compounds were validated for a kinase target in oncology and 141 top-ranked compounds were validated for a non-kinase enzyme in a CNS indication. Both assays exhibited a very high predictive performance of ROC-AUC > 0.9. Simm and colleagues reported a 50-fold and 280-fold increase in the hit rate of their validation compared to the HTS hit rate[6]. It is worth noting that sampling a larger fraction of compounds imposes limits on the maximum achievable enrichment with the bootstrapping approach. For instance, by sampling the top 5% the maximum achievable enrichment rate is 20-fold. Conversely, sampling only the top 0.5% the maximum achievable enrichment rate would be 200-fold.

While phenotypic-based bioactivity prediction requires in vitro data, such datasets do not need to be expensive to generate; the Cell Painting assay protocol was designed to be rich in information while low-cost to perform[9]. Notably, we demonstrate the direct prediction of bioactivity from brightfield images. This successful prediction from brightfield images underscores the substantial phenotypic information related to compound perturbations present in these images. Consequently, this implies that fluorescent images may not be a prerequisite, thus reducing the need for staining reagents and the use of advanced microscopes. This broadens the applicability of our methodology and holds the potential for a significant reduction in implementation costs. In a drug discovery setting, the initial expenses for a single Cell Painting HTS would be easily recouped, as traditional HTS assays could be reduced and replaced by bioactivity predictors using the phenotypic data generated, combined with smaller, focused screens. We provide a comprehensive analysis and guidance on which assays, technologies and targets are especially suited for phenotypic bioactivity prediction. Notably, kinase targets and cell-based assays exhibited strong performance, and additional trends can be recognized albeit at small sample size (Fig. 3).

In summary, we have shown that phenotypic screening data combined with readily available single concentration data can be used for bioactivity prediction in a wide range of assays, with high performance across different target classes, assay technologies, and disease areas. Beyond the use of fluorescence data, we have also shown that brightfield data can reach a performance level comparable to or better than structure-based predictors, highlighting that staining the cells might not be necessary, but that it does provide slightly better predictions. Overall, the results paint a positive picture for phenotypic-based bioactivity models to complement structure-based predictors, where data can be generated in a cost-effective manner and with several use-cases.

## Methods

### Generation of the Cell Painting and HTS dataset
A set of 8,300 compounds was selected and screened in the Cell Painting assay. The compounds set was selected based on chemical diversity, known annotations, compound availability, and on general representation in historical HTS screens.

**Cell Painting.** The Cell Painting staining and imaging procedure was performed according to the protocol by Bray et al.[9] with some adjustments to stain concentrations and methodology as described recently[18,35]. Briefly, U-2 OS cells (ATCC Cat# HTB-96) were cultured in McCoy's 5 A media (Fisher Scientific, #26600023) with 10% (v/v) fetal bovine serum (Fisher Scientific, #10270106) at 37 °C, 5% (v/v) CO2, 95% humidity. Cells were seeded in CellCarrier-384 Ultra microplates (Perkin Elmer, #6057300) at 1500 cells per well 24 h prior to compound addition. Cells were treated with compounds at 10 μM for 48 h and then stained with 500 nM MitoTracker (ThermoFisher, M22426) for 30 minutes. Cells were then fixed with 3.2% v/v formaldehyde in PBS for 20 minutes and washed using a BlueWasher centrifugal plate washer (BlueCat Bio, Neudrossenfeld, Germany). Cells were

permeabilized for 20 min at room temperature using 30 µL of 0.1% (v/v) Triton X-100 in HBSS (Sigma Aldrich, #T8787). Following this, 15 µL of a stain solution was applied, consisting of 5 µg/mL Hoechst 33342 (ThermoFisher, H3570), 1.5 µg/mL Wheat-germ Agglutinin Alexa Fluor 555 conjugate (ThermoFisher, W32464), 10 µg/mL ConcanavalinA Alexa Fluor 488 conjugate (ThermoFisher, C11252), 5 µL/mL Phalloidin Alexa Fluor 568 conjugate (ThermoFisher, A12380), and 9 µM SYTO14 (ThermoFisher, S7576) to each well. After 30 min incubation cells were washed with HBSS. The plates were sealed and subsequently subjected to imaging.

Cell imaging was conducted using a CellVoyager CV8000 (Yokogawa, Tokyo, Japan) equipped with a 20× water-immersion objective lens (Olympus, Tokyo, Japan; NA 1.0). Five channels were employed to visualize the various fluorescent stains: DNA (excitation: 405 nm; emission: 445/45 nm), ER (excitation: 488 nm; emission: 525/50 nm), RNA (excitation: 488 nm; emission: 600/37 nm), AGP (excitation: 561 nm; emission: 600/37 nm), and Mito (excitation: 640 nm; emission: 676/29 nm). Brightfield images were obtained at 3 focal planes of 4 µm distance. Four fields of view were captured per well.

**Activity data.** Corresponding activity data were extracted from an internal HTS assay database. Historical HTS data were annotated as binary active/inactive using thresholds individually determined for each assay during assay development. Data were not available for all the selected compounds in all available historical screens, and we set a threshold of a minimum number of 50 active and 50 inactive datapoints required for a screen to be included. The included screens covered a diverse range of assay technologies, target types, therapeutical areas, etc. Following these criteria, the resulting dataset included around 70,000 images, covering 8,300 compounds with associated activity data in 140 unique assays. The label matrix had a 47.8% fill rate and an average of 3% of the known compounds labeled as active.

### JUMP consortium and ChEMBL datasets

**Cell Painting Images.** Publicly available Cell Painting data was collected from the JUMP consortium dataset[13], CPG0016 available from the Cell Painting Gallery on the Registry of Open Data on AWS. The compounds subset of the dataset contains Cell Painting data for over 115,000 unique compounds. These have been imaged following the protocol described in[13], using U-2 OS cells, treated with compounds at 10 µM. Compound data from source 11 was selected as it contained the largest overlapping subset of data with sufficient activity data, see following subsection of activity data information.

**ChEMBL activity data.** Publicly available activity data was collected from ChEMBL[14]. The compounds found in the JUMP-CP dataset were cross-referenced with those available with activity Potency readouts in the ChEMBL database (version 33). Using the activity flag field binary labels were assigned for each compound-assay datapoint. With [active] and [Active] treated as active and [inactive] and [Inactive] as inactive. The majority of overlapping compounds were found to come from source 11 of the JUMP-CP dataset and to reduce the impact of varying imaging settings between sources, data from other sources were disregarded. Following this, only assays with enough activity data among the remaining compounds were kept (more than 50 active and inactive compounds). Amounting to 10,660 in 29 distinct assays. To access the dataset, we provide a script along with a comprehensive step-by-step guide for the automated download and pre-processing, available at https://github.com/cfredinh/bioactive

### Bio-activity prediction setup and evaluation

**Data splits and cross-validation.** Using a cross-validation setup, we split the data into 6 different folds with each compound only included in one-fold. For each cross-validation setting 4 splits were used as training, 1 for validation and 1 left out as test. The model and hyperparameter selection were done using only the training and validation splits, while the test set was only used to report final performance.

The distribution of compounds between different splits was done based on structural similarity.

Using RDKit[36], all compound SMILES representations were converted to ECFP4 1024 Bit. Using RDKit Butina ClusterData function, the ECFP4 representations was used to group the data into unique clusters. These clusters were divided into 6 unique folds of similar size such that structurally similar compounds, belonged to the same fold.

**Prediction setup.** Depending on the input modality, different Machine Learning models were used. Multi-Layer Perceptrons (MLPs) were used for the cell-feature-based model and the structure-based, described below in section 'Cell-feature model' and section 'Structure fingerprints model' For the Fluorescence and Brightfield images ResNet50s were used.

All models were trained to predict if a compound was active or inactive in each of the unique 140 assays as a multi-label binary prediction task. All networks were trained with 140 output neurons representing the 140 unique assays. A sigmoid activation function was used to normalize the range of values for each output neuron individually to the range of [0,1].

The models were trained with Binary Cross-Entropy combined with Focal Loss[37]. Both propagate a loss signal from each of the output neurons of the network. Given the fact that not all compounds have been tested in all assays, the label matrix is incomplete. Thus, the activity of many of the compounds are unknown and no loss signal backpropagated from those neurons.

Area Under the Receiver Operating Characteristic Curve (ROC-AUC) was used to evaluate the model's ability to separate the actives from inactive. Since many activity labels are missing, the performance is only calculated for compounds with known activity values.

The final performance is calculated based on the per-compound averaged prediction, where the output predictions for each image of the same compound is averaged using the mean. We report both the mean ROC-AUC over all assays as well as the individual ones.

### Approach

**Fluorescence image-based model.** Fluorescent microscopy images were stored as 16-bit TIFFs of size 1992×1992. The images were preprocessed and normalized such that the top and bottom 1 percentile intensity values were clipped for each image to remove noise and outliers.

Before being sent to the network as input during training, the images were augmented, including spatial down sampling, z-normalization, random cropping, horizontal and vertical flipping, random 90-degree rotations and color shifting.

A ResNet50[10] model was used as a feature extractor, using 448×448 pixel image crops as input. Transfer learning was done by utilizing a ImageNet pre-trained network, downloaded from torch-hub see He et al.[10] for details regarding the pre-training.

The network was adapted to allow 5 channel input images by adding two channels to the input convolutional filter, done by repeating the two first channels of each convolutional filter (Supplementary Fig. 6). The linear layer of the pre-trained model was replaced with a re-initialized one with 140 output neurons to match the number of assays.

All models were trained on two NVIDIA-Tesla 32 Gb GPUs, using pytorch DDP[38]. A hyper-parameter search was performed using nested cross-validation in each one of the cross-validation splits, using three splits for training, one for nested validation and one for nested testing. Searching for optimal learning rate, weight decay, and optimizer. The identified hyper-parameters were then used to train the pre-trained ResNet50 for 100 epochs, using early stopping based on the validation

ROC-AUC performance and learning rate stopping on plateau. The best identified parameters were learning Rate 0.2/256, Optimizer SGD, weight decay $10^{-4}$. A model was trained for each of the cross-validation splits, with the best performing model checkpoint based on validation-set performance being used to predict the likelihood of activity for the respective test set split.

The model used on the publicly available data followed the same steps, with some minor changes to adapt to some slight differences in data. The images are of size 1080×1080 pixels and Pre-Processing done with the *DeepProfiler* package[39]. Once again, the hyper-parameter tuning was performed using the same setup as described above. Searching for optimal learning rate, weight decay, and optimizer. The best performing settings were learning rate $1e^{-3}$, weight decay $1e^{-4}$ and SGD. See Supplementary Fig. 7 for loss curves.

**Brightfield image-based model.** Brightfield microscopy images were stored as 16-bit TIFFs of size 1992×1992 with three images captured at different focal planes at each site (+/− 4 μm around the central Z plane). The images were pre-processed and normalized such that the top and bottom 1 percentile values were clipped for each image to remove noise and intensity outliers.

The Brightfield image-based model was trained following the same procedure as for the fluorescent image-based model, although the default setting of three channels as input was used, stacking the three focal planes into one image. Hyper-parameter tuning and evaluation were done following the same procedure as for the fluorescent images. The best identified parameters were the same across splits learning rate 0.2/256, Optimizer SGD, weight decay $10^{-4}$.

**Cell-feature model.** Single-cell image features were extracted for each plate using the Columbus software package (v 2.9.1, PerkinElmer). Features such as nucleus size, cell radius, average nuclei intensity, were extracted. In total 1,176 features were used for each cell. This was done for each cell and averaged per well. The averaged data were normalized by z-normalization per plate using DMSO controls using feature-wise median and median absolute deviation. Features with a variance below 1.0 were deemed uninformative and removed.

Following feature normalization and removal, the remaining features were used as model input. A Multi-layer perceptron (MLP) was used for prediction. Like the previous two model types, binary-cross entropy combined with focal loss was used to train the model.

Due to the more manageable size and computational requirements of the feature-based model, a larger hyper-parameter tuning was done using nested cross-validation in each of the cross-validation splits. Searching for optimal, model-depth, layer-width, weight-decay, learning rate and optimizer. The identified hyper-parameters were then used to train MLP for 150 epochs, using early stopping based on the validation ROC-AUC performance and learning rate dropping on plateau. The best identified parameters were 3-hidden layers, 1024-layer width, weight decay 0.0, learning rate 5.0, SGD. A model was trained for each of the cross-validation splits with the best performing model checkpoint, based on validation-set performance, being used to predict the likelihood of activity for the respective test set split.

**Structure fingerprints model.** Each of the 8,300 compounds were represented using the commonly employed Extended Connectivity Fingerprints (ECFP) with a compound component diameter of 4. Using RDKit[36], all compound SMILES[40] representations were converted to ECFP4. The ECFP4 representation was done using 1024-dimensional bit string, which was used as input to the activity prediction model.

A Multi-layer perceptron (MLP) was used for the predictions, using the 1024-dimensional ECFP4 representation as input and predicting activity in each of the 140 assays as output. Binary-cross entropy combined with focal loss was used to train the model.

Like the Cell-feature model, a full hyperparameter tuning was done using nested cross-validation in each of the cross-validation splits. Searching for optimal, model-depth number of hidden layers, layer-width, weight-decay and learning rate. The identified hyper-parameters were then used to train MLP for 150 epochs, using early stopping based on the validation ROC-AUC performance and learning rate dropping on plateau. The best identified parameters were (3-hidden layers, 512-layer width, weight decay 0.0, learning rate 2.0). A model was trained for each of the cross-validation splits with the best performing model checkpoint, based on validation-set performance, being used to predict the likelihood of activity for the respective test set split.

## Statistics and reproducibility

Performance variations between model types were analysed using Friedman rank sum test, using the assays as blocking factors. This was calculated using Friedman-Chi-Square test using SciPy[41] stats package followed by Nemenyi's-Friedman post-hoc test.

Performance differences depending on assay characteristics was analysed using one-way ANOVA with post-hoc tests, calculated using Kruskal-Wallis test in SciPy stats package[41], followed by Conover pair-wise test to determine if there were any statistically significant differences between sub-groups.

The sample size used was not determined based on any statistical method, all data available was included. No data were excluded from the analysis and the investigators were not blinded.

## Diversity evaluation

Tanimoto similarity[42] between the ECFP4 fingerprints of compounds was used to determine how structurally similar each compound pair was. To assess the diversity of the top ranked compounds according to each predictive model, the top 20 ranked compounds in each test set were compared to the known actives in their respective training set. Each top ranked compound was compared to all the known actives and the most similar one was identified for each of the 20 compounds, meaning the one with highest Tanimoto Similarity was then assigned as the most similar.

## Follow-up screening and calculation of enrichment

Top ranked compounds in four of the assays were selected for follow-up validation in secondary screening. These compounds were selected based on their activity scores in the test set. This allowed us to select the top compounds from the full dataset without data leakage. The Fluorescence Whole Image based model type was used to assign activity scores for each compound. The top ranked 5% of compounds were randomly sampled for each of the four follow-up assays, with varying numbers of compounds selected for each of the assays. In addition, a random subset of compounds was also sampled for follow-up screening and used to calculate ROC-AUC metrics in the follow-up assays.

In each of the available follow-up assays a baseline estimate of assay activity was established by probing at least 500 randomly sampled compounds. This gives an estimate of the overall hit rate in each assay.

The likelihood of activity for each of the compounds in all six test-splits were combined and ranked together, based on the prediction of the Cell Painting Fluorescent Whole Image ResNet50. For each follow-up assay, the top 5% of compounds deemed most likely to be active in the corresponding HTS assay, were randomly sampled. These combined with the randomly sampled set for each of the assays, were screened in their respective follow-up assays.

The ROC-AUC values in the follow-up screen were then evaluated using the randomly sampled compounds. The randomly sampled compounds were also used to assess the baseline hit rate for each of the assays, which was used for the enrichment analysis. The

enrichments at different percentiles were then calculated using bootstrapping of the activity values of the compounds above that percentile. See Supplementary Table 1 for further details.

## Reporting summary

Further information on research design is available in the Nature Portfolio Reporting Summary linked to this article.

## Data availability

The raw HTS datasets generated and analysed in this study are protected and are not available due to them being AstraZeneca proprietary information. The publicly available Cell Painting data used in this study are available from the JUMP consortium dataset[13], CPG0016 available from the Cell Painting Gallery on the Registry of Open Data on AWS. (https://registry.opendata.aws/cellpainting-gallery/). The compound activity data used in this study are available from the ChEMBL[14] database version 33 (https://www.ebi.ac.uk/chembl/). We provide a script along with a comprehensive step-by-step guide for the automated download and pre-processing of the Cell Painting/ChEMBL dataset, available at https://github.com/cfredinh/bioactive. Source data are provided as a Source Data file. Source data are provided with this paper.

## Code availability

The code is available on github at https://github.com/cfredinh/bioactive.

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

## Acknowledgements

The authors would like to thank Guy Williams, Diane Smith, Hannah Semple, and Elizabeth Mouchet for their invaluable help in producing the Cell painting dataset used in this publication. We would like to thank the AZ SCP team for assisting with computational resources. We thank Eva Hansson, Craig Hughes, Linda Fredlund, Fredrik Wågberg, Nancy Su, Vijay Chandrasekar, and Xiang Zhai for their assistance in generating the imaging dataset and or running wet-lab experiments. J.F-H is funded by the Wallenberg AI, Autonomous Systems and Software Program (WASP) and AstraZeneca.

## Author contributions

J.F-H., J.K. C.-H.L., KJ.L., R.T., E.M. and K.S. performed study concept and design, J.F-H. performed the computational implementation and analysis. KJ.L. and E.M. conducted wet-lab work. All the authors contributed to the writing of the paper and have read and approved the current version of the paper.

## Competing interests

J.F-H., KJ.L., R.T., J.K., C.-H.L. and E.M. were all employees of AstraZeneca plc at the time of this work. AstraZeneca provided funding and support for this research in the form of salaries for the authors, the HTS dataset, computing resources, experimental reagents, and experimental support, but did not have any additional role in the study design, data collection and analysis, decision to publish, or preparation of the manuscript. K.S. has no competing interests to declare.
