## [Peer Review File · Nature Communications]

Reviewers' Comments:

Reviewer #1:

Remarks to the Author:

Summary: The authors present a study in which they predict HTS screening data based on high-content images.

General comments: The work lacks novelty and is almost identical with two prior studies. The presented approach is technically mostly sound. Predictive modeling of bioactivities of small molecules and representation learning of high-content imaging is a highly relevant topic. The clarity of writing is lacking because of imprecise terminology.

Reproducibility: Neither code nor data is available, which means that this work is not reproducible.

Positive aspects:

- Correct comparison via nested cross-validation and scaffold-splitting
- Statistical testing between methods
- Relatively well embedded into scientific literature and related works

Major comments:

1) The authors cite two studies [4] and [8] (in their reference list), that are almost identical to the presented study: [4] shows that screening data (including HTS) can be predicted based on high-content images, and [8] includes CNNs, e.g. ResNets, in the study. The authors claim that their "result extend this by showing the capacity to learn from more readily [...] activity readouts", but these types of readouts are already included in the data sets of [4] and [8]. The earlier studies [4] and [8] were even much larger, in terms of compounds and classification tasks, such that this study is even a step backward. The authors should present a novel method or approach.

2) Technically, the approach appears sound. The data has been split into training, validation and testing, and evaluated via nested cross-validation (please clearly state this). The only thing that is lacking is that the hyperparameter selection process is not well explained. The authors should clearly describe how they select the hyperparameters of each method. Additionally, details on the chemistry-based method would be important!

3) The relevance of the principle modeling problem, which is prediction of bioactivities based on high-content imaging data, is highly relevant. However, the presented study is essentially the same as the prior studies, their hardly is any added value for the scientific community.

4) Clarity and writing. The terminology used is imprecise and many technical terms are used but not introduced, e.g. "initial hit finding", "property prediction" vs "bioactivity prediction", "phenotypic profile", etc. In general, lots of terms from different scientific communities are introduced, used, and often not well-explained or not defined. In that regard, the formulations are very imprecise. The authors should improve the terminology used in this work to improve the precision and clarity of writing.

Minor comments:

- Performance values within the text (and in tables) should have error bars or confidence intervals. The authors could easily provide this, since they have made several re-runs in their cross-validation procedure.

- In contrast to prior belief, that predictions based on chemical structure are predictively better than those based on imaging data, the results show that predictive performance of imaging-based methods is better. Could the authors provide more insights on that?

- Discussion section "Self-Supervised Learning": this is already being done, e.g. see reference [1]

References:

[1] Sanchez-Fernandez, et al. "Contrastive learning of image-and structure-based representations in drug discovery." In ICLR2022 Machine Learning for Drug Discovery. 2022.

Reviewer #2:

Remarks to the Author:

Haslum and coauthors investigate the use of deep learning in conjunction with Cell Painting to predict bioactivity of compounds. Utilizing Cell Painting data and preliminary single-concentration activity outputs from high-throughput screening (HTS) assays, researchers studied the predictability of compound activity across 140 unique assays. The models demonstrated high predictive performance, particularly in cell-based assays and kinase targets, with an average ROC-AUC of 0.744. This approach outperformed conventional structure-based predictions in terms of predictive performance and compound structure diversity. The results suggest these models can predict compound activity in a range of HTS assays.

I like the quality and scientific rigor of the research presented in this paper. The comprehensive analysis and presentation of the results demonstrate a high level of scientific excellence, which is commendable.

The approach of integrating deep learning with cell painting assays for bioactivity prediction is highly relevant, providing an insightful contribution to early drug discovery studies. The ability to predict compound activity using morphological profiles indeed has the potential to significantly streamline the drug discovery process.

However, I would like to propose a few suggestions (not in ranked order) to further enhance the quality and accessibility of this work:

1. Source code availability: While it is understood that the source code can be obtained from the authors upon request, it would greatly benefit the wider scientific community if the code was made publicly accessible. This would foster transparency and reproducibility, and also encourage collaborations and further improvements.
2. Detailed Cell Painting Screen description: The description of the Cell Painting Screen could be expanded upon. Providing more information on the type of cells used, the duration of compound exposure, and the concentration administered would increase understanding and allow for better replication of the study.
3. Description of ResNet-50 pretraining: The paper could benefit from a more comprehensive description of the pretraining process of the ResNet-50 model. Further details would enable readers to understand better the fine-tuning performed and its impact on model performance.
4. Learning curves for all models: Including learning curves demonstrating the loss during training and testing for each of the four models would provide valuable insight into their respective performance and behavior over time.
5. Visualization of learned filters: It would be interesting and informative to see some images of the filters learned in the first convolutional layer. This would provide a deeper understanding of the transformations the model is learning and how these contribute to its predictive abilities.

6. Detailed Inter-assay correlation: I would suggest a more in-depth presentation of the inter-assay correlation. Including pairwise scatter plots, for example, could provide a clearer visual representation and further enhance understanding of these correlations.

Reviewer #1 (Remarks to the Author):

Summary: The authors present a study in which they predict HTS screening data based on high-content images.

General comments: The work lacks novelty and is almost identical with two prior studies. The presented approach is technically mostly sound. Predictive modeling of bioactivities of small molecules and representation learning of high-content imaging is a highly relevant topic. The clarity of writing is lacking because of imprecise terminology.

Reproducibility: Neither code nor data is available, which means that this work is not reproducible.

Positive aspects:

- Correct comparison via nested cross-validation and scaffold-splitting
- Statistical testing between methods
- Relatively well embedded into scientific literature and related works

Thank you for your review and constructive feedback. We have thoroughly revised the manuscript based on the suggestions. First, to substantiate our method and to allow for better reproducibility, we have included results from applying our model to a publicly available dataset (JUMP-CP imaging data and ChEMBL activity data). Second, we have taken the necessary measures to make our code publicly available. Link to github: <https://github.com/cfredinh/bioactive> Third, we have extensively revised the manuscript to improve clarity and conciseness, with a focus to more effectively convey the contribution of our work and its distinctive aspects compared to existing literature. Finally, we have addressed all your comments as well as the editors and the other reviewers' comments, with the hope that these revisions address your initial concerns.

Major comments:

1) The authors cite two studies [4] and [8] (in their reference list), that are almost identical to the presented study: [4] shows that screening data (including HTS) can be predicted based on high-content images, and [8] includes CNNs, e.g. ResNets, in the study. The authors claim that their "result extend this by showing the capacity to learn from more readily [...] activity readouts", but these types of readouts are already included in the data sets of [4] and [8].

We recognize that our previous communication may not have effectively conveyed the distinctive aspects of our work compared to existing literature. While we acknowledge that our technical approach is not entirely novel, the uniqueness of our contribution lies in several key facets:

- **Input Data:** *Unlike prior research, we demonstrate the predictive power of utilizing real-world, noisy, single-point activity data. Our results reveal that, for many assays, even a small number of single concentration data points—sometimes just a few hundred—are sufficient for our model to effectively learn and predict the entire library's activity in that specific assay. This has the potential to drastically reduce the number of compounds that need to be screened.*
- **Imaging Modality:** *We extend the scope of accurate predictions beyond fluorescent images to include brightfield images, broadening the applicability and potentially drastically reducing implementation costs of our methodology.*
- **Validation:** *Our work goes beyond in-silico predictions, as we rigorously validate our findings through in-vitro experiments.*
- **Comprehensive Analysis:** *To fortify the practicality of our approach, we conduct a thorough analysis. Our results show that Cell Painting-based bioactivity prediction exhibits robustness across various assay types, technologies, and target classes. Additionally, we identify specific*

REBUTTAL LETTER - MANUSCRIPT NCOMMS-23-14357

targets and assays that are particularly well-suited for bioactivity prediction. This comprehensive analysis equips readers with the knowledge and tools needed to implement our approach in practice.

We have extensively revised large sections of the manuscript to explain more clearly what your approach adds over previous approaches e.g., in the abstract lines 15-32, in the introduction lines 80-88, lines 94-98, lines 104-107 & lines 110-112; in the results lines 249-253; and in the discussion lines 353-357, lines 378-383 & lines 417-421.

The earlier studies [4] and [8] were even much larger, in terms of compounds and classification tasks, such that this study is even a step backward. The authors should present a novel method or approach.

We see it as one of the key differences that the studies mentioned did use larger sets of data. The previous studies employed dose-response data and/or multiple data points per compound, and several EC50 and IC50 cutoffs, effectively replicating the same assay. The goal of our study was to implement deep learning based bioactivity prediction in a practical context, emphasizing the reduction of screening campaign sizes and the potential for primary screening with more complex assays. The use of dose-response data, or gathering multiple datapoints per compound, would have contradicted the fundamental objectives of our approach. While larger datasets may offer enhanced statistical robustness, we demonstrate that smaller datasets are not only sufficient but also directly applicable and practical.

2) Technically, the approach appears sound. The data has been split into training, validation and testing, and evaluated via nested cross-validation (please clearly state this). The only thing that is lacking is that the hyperparameter selection process is not well explained. The authors should clearly describe how they select the hyperparameters of each method.

Thank you for pointing this out. Nested cross-validation was indeed used and details to further describe this have now been added to the Methods sections, see lines 505-509, 550-555, 572-574, 586-594 and 605-612. In the revised version we state the use of nested cross-validation in the results section page 3, line 139.

The hyperparameters were selected based on the performance in the nested cross-validation. The hyperparameter selection process for the ResNet50 based models was performed for the weight-decay, learning rate, and optimizers. The Cell-feature and structure-based model used Multi-Layer perceptron models and more hyperparameters were searched for. Including model-depth, layer-width, weight-decay, learning rate and optimizer. We now explain this in detail in the updated text page 15, lines 586-594 and 605-612.

Additionally, details on the chemistry-based method would be important!

The structure-based model relies on commonly used RDKit calculated ECFP4 1024 bit, representations of each compound. The 1024-bit vector was used as input to a Multi-Layer perceptron network, with 140 outputs representing each assay. The same binary cross-entropy loss combined with focal loss used for the other input modalities was also used in this case. Once again nested cross-validation was performed to select the best hyperparameters as described in the previous paragraph. Additional details on the chemistry-based method have been added to the text lines 596-612.

3) The relevance of the principle modeling problem, which is predictign bioactivities based on high-

REBUTTAL LETTER - MANUSCRIPT NCOMMS-23-14357

content imaging data, is highly relevant. However, the presented study is essentially the same as the prior studies, their hardly is any added value for the scientific community.

Based on your valuable feedback and that of the other reviewer, we have taken steps to further strengthen the manuscript address the clarity and visibility of our work's novelty and potential impact to differentiate from previous publications. To reinforce our method, enhance reproducibility, and provide additional value for the scientific community we provide our code and have incorporated results from applying our model to a publicly accessible dataset (JUMP-CP imaging data and ChEMBL activity data). In addition, we have made comprehensive revisions to the manuscript based on the feedback received to communicate our work's contributions and highlight its distinctive aspects in comparison to existing literature more effectively.

Please also see our reply to your major comment 1).

4) Clarity and writing. The terminology used is imprecise and many technical terms are used but not introduced, e.g. "initial hit finding", "property prediction" vs "bioactivity prediction", "phenotypic profile", etc. In general, lots of terms from different scientific communities are introduced, used, and often not well-explained or not defined. In that regard, the formulations are very imprecise. The authors should improve the terminology used in this work to improve the precision and clarity of writing.

We acknowledge that certain technical terms seem clear to us authors, but can be imprecisely defined, or could mean something different to readers from different fields. Thus, thank you for pointing this out. We have revised the manuscript accordingly and introduced or revised respective technical terms where they first occur in the text e.g., "initial hit finding" revised to "hit finding" lines 40, 45, 54, 350 & 352; "SAR models" defined in lines 62-64; "property prediction" defined in lines 63-64 & 73; "bioactivity prediction" defined in line 63-64; "phenotypic profile" defined in line 67-72.

Minor comments:

- Performance values within the text (and in tables) should have error bars or confidence intervals. The authors could easily provide this, since they have made several re-runs in their cross-validation procedure.

Thank you for pointing this out, we have now added standard deviation values where applicable throughout the text e.g., line 21, 145, 160, 205-208, 290.

- In contrast to prior belief, that predictions based on chemical structure are predictively better than those based on imaging data, the results show that predictive performance of imaging-based methods is better. Could the authors provide more insights on that?

To the best of our knowledge, there has been limited work comparing models based on chemical structure and those based on imaging data (REF [10.1021/acs.jcim.0c00864](https://doi.org/10.1021/acs.jcim.0c00864)). Concurrent research has shown that image-based model performance is often comparable (REF [10.1186/s13321-023-00723-x](https://doi.org/10.1186/s13321-023-00723-x)) or slightly superior (REF [10.1038/s41467-023-37570-1](https://doi.org/10.1038/s41467-023-37570-1)), aligning with our own observations.

Several factors appear to influence these models' performance and the comparison between input modalities. These factors include the data splitting strategy, the information density of the modality, and the specific assays used.

One significant factor involves how the dataset is divided into training, validation, and test sets (REF [10.1007/s11224-011-9757-4](https://doi.org/10.1007/s11224-011-9757-4)) A purely random split of compounds can yield an overly

REBUTTAL LETTER - MANUSCRIPT NCOMMS-23-14357

optimistic performance estimate for structure-based approaches. This is because it fails to simulate the predictor's performance in unknown areas of the structural space (REF 10.1038/nrd4128). In contrast, structure or scaffold-based splitting eliminates overlap between similar compounds in data splits, making the task more challenging for structure-based models and better assessing performance in uncharted structural regions. We employed this approach, even though it may seem to disadvantage structure-based models. This method is commonly used because it provides a more accurate estimate of performance, and identifying active compounds highly similar to those in the training set has limited practical utility, as diverse compounds are of greater interest.

Beyond the impact of data splitting, performance differences can also be attributed to variations across different assay and target types. As observed previously, performance varies from assay to assay. We also note that in certain assays, the structure-based model outperforms the image-based model, as illustrated in Supplementary Figure 7. However, our analysis indicates that, on average, image-based models outperform structure-based models. Notably, our study highlights the performance differences among various assay and target types in Figure 3.a. It demonstrates that the performance contrast between structure and image-based assays is not significant when studying biochemical assays. However, in the case of cell-based assays, image-based models significantly outperform their structure-based counterparts. Given that a considerable portion of the assays in this study are cell-based, image-based models, on average, outperform structure-based ones. Depending on the proportion of assay types, performance may be comparable or inverted. This analysis of performance variations in different target and assay types provides valuable new insights and is one of the novel aspects of our work.

We discuss this in the revised manuscript lines 387-397.

- Discussion section "Self-Supervised Learning": this is already being done, e.g. see reference [1]
References:

[1] Sanchez-Fernandez, et al. "Contrastive learning of image-and structure-based representations in drug discovery." In ICLR2022 Machine Learning for Drug Discovery. 2022.

Thank you for highlighting this work, we were not aware of this work at the time of submission, we have included the reference and discuss Self-Supervised learning among other approaches in lines 399-409.

Reviewer #2 (Remarks to the Author):

Haslum and coauthors investigate the use of deep learning in conjunction with Cell Painting to predict bioactivity of compounds. Utilizing Cell Painting data and preliminary single-concentration activity outputs from high-throughput screening (HTS) assays, researchers studied the predictability of compound activity across 140 unique assays. The models demonstrated high predictive performance, particularly in cell-based assays and kinase targets, with an average ROC-AUC of 0.744. This approach outperformed conventional structure-based predictions in terms of predictive performance and compound structure diversity. The results suggest these models can predict compound activity in a range of HTS assays.

I like the quality and scientific rigor of the research presented in this paper. The comprehensive analysis and presentation of the results demonstrate a high level of scientific excellence, which is commendable.

Thank you very much.

The approach of integrating deep learning with cell painting assays for bioactivity prediction is highly relevant, providing an insightful contribution to early drug discovery studies. The ability to predict compound activity using morphological profiles indeed has the potential to significantly streamline the drug discovery process.

However, I would like to propose a few suggestions (not in ranked order) to further enhance the quality and accessibility of this work:

1. Source code availability: While it is understood that the source code can be obtained from the authors upon request, it would greatly benefit the wider scientific community if the code was made publicly accessible. This would foster transparency and reproducibility, and also encourage collaborations and further improvements.

We agree and have now taken the necessary measures to make the code publicly available. Link to github: <https://github.com/cfredinh/bioactive> Further, while the AZ internal data set used in this work cannot be published, we have now applied the same predictive framework to a publicly available dataset as a proof of concept. While the setting is not a one-to-one correspondent to the internal data, this allows for easier reproducibility and transparency. Please see lines 157-163, Figure 1c, lines 480-500, lines 559-563.

2. Detailed Cell Painting Screen description: The description of the Cell Painting Screen could be expanded upon. Providing more information on the type of cells used, the duration of compound exposure, and the concentration administered would increase understanding and allow for better replication of the study.

We have expanded the description of the Cell Painting method including all information on cells, materials, staining method, and imaging conditions in lines 442 to 464.

3. Description of ResNet-50 pretraining: The paper could benefit from a more comprehensive description of the pretraining process of the ResNet-50 model. Further details would enable readers to understand better the fine-tuning performed and its impact on model performance.

Thank you for pointing this out, we have now added more details on this in lines 542-544. The ResNet50 network was pre-trained on the ImageNet 1k dataset 10.1109/CVPR.2009.5206848,

REBUTTAL LETTER - MANUSCRIPT NCOMMS-23-14357

the training procedure can be found in 10.1109/CVPR.2016.90 We load the pre-trained model from the Torch Model hub. This model is used as a starting point of the training, relying on transfer learning for better initial features. This decision was made based on several previous works showing the benefits of transfer learning from ImageNet even when transferring to widely different domains.

4. Learning curves for all models: Including learning curves demonstrating the loss during training and testing for each of the four models would provide valuable insight into their respective performance and behavior over time.

To provide additional insight into the performance behaviour and reproducibility of our approach, we have run our model on a public dataset (figure 1c, lines 480-500, lines 559-563). We now provide learning curves demonstrating the loss during training and testing in supplementary figure 8.

5. Visualization of learned filters: It would be interesting and informative to see some images of the filters learned in the first convolutional layer. This would provide a deeper understanding of the transformations the model is learning and how these contribute to its predictive abilities.

We have now added a visualize first convolutional filters in Supplementary Figure 8. The ResNet50 model was initialized using ImageNet pre-trained weights, employing transfer learning to utilize features learnt in the natural domain. As expected, when employing Transfer Learning in a low-to-mid sized image dataset, the initial feature extractors weights are largely maintained and the Gabor-filter like patterns that are transferred can be clearly seen.

6. Detailed Inter-assay correlation: I would suggest a more in-depth presentation of the inter-assay correlation. Including pairwise scatter plots, for example, could provide a clearer visual representation and further enhance understanding of these correlations.

Thank you for suggesting this interesting analysis. We have now added Cluster map plots featured in supplementary figure 10 and discussed in lines 147-149, to visualize the inter-assay correlation among our predictions. While some assays exhibited low correlation, others displayed strong positive correlations, while still others showcased negative correlations. Extensive investigation into these correlations revealed that they did not exhibit any discernible associations with predictive performance, assay type, technology, or target type.

Reviewers' Comments:

Reviewer #1:

Remarks to the Author:

Summary: The authors present a study in which they predict HTS screening data based on high-content images.

General comments: The authors were only able to accommodate a few minor points. The main concern, which is lack of novelty, remains and my concerns have even aggravated.

Major comments:

1) Lack of novelty: The presented study is still highly similar to reference [4] and [8]. In their rebuttal, the authors now point out the following four potential aspects of novelty:

a) Noisy single-point activity data sufficient (a few hundreds of points): This has been already shown in prior works (e.g. [4] and [8]) that this is sufficient. In current state-of-the-art methods, it is shown that it is possible for few-shot (less than 10 per activity class) or zero-shot [1] setting.

b) Imaging modality: the authors mention that they include bright field microscopy images. While this is an interesting observation, this type of transfer from pre-training on one modality (e.g. natural images) and then transferring to new modalities (e.g. biomedical images) is well-known and studied since the advent of deep convolutional networks ([2],[3]).

c) Validation in in-vitro experiments: has also been done in [4].

d) Comprehensive analysis: this is indeed slightly novel, but not a radically new finding. QSAR models are often analyzed in this way. The authors should focus on a single, clear and relevant novel aspect of their work and write the paper with this novelty as main theme of the paper.

2) The study is smaller than previous studies with only 140 assays analyzed while [4] already modelled ~1200. The authors should benchmark their method on previously established datasets and benchmarks, concretely the benchmark proposed in [8] and [1]. The authors should use datasets within a size of state-of-the-art methods.

3) Reproducibility: has improved, but is still low because the pre-processed dataset has not been provided. The authors should provide the pre-processed dataset in simple downloadable and accessible form, such that other researchers can easily reproduce their findings.

References:

- [1] Sanchez-Fernandez, A., Rumetshofer, E., Hochreiter, S., & Klambauer, G. (2023). CLOOME: contrastive learning unlocks bioimaging databases for queries with chemical structures. *Nature Communications*, 14(1), 7339.
- [2] Yu, Y., Lin, H., Meng, J., Wei, X., Guo, H., & Zhao, Z. (2017). Deep transfer learning for modality classification of medical images. *Information*, 8(3), 91.
- [3] Kim, H. E., Cosa-Linan, A., Santhanam, N., Jannesari, M., Maros, M. E., & Ganslandt, T. (2022). Transfer learning for medical image classification: a literature review. *BMC medical imaging*, 22(1), 69.
- [4] Simm J, et al. Repurposing High-Throughput Image Assays Enables Biological Activity Prediction for Drug Discovery. *Cell Chem Biol* 25, 611-618 e613 (2018).
- [8] Hofmarcher M, Rumetshofer E, Clevert DA, Hochreiter S, Klambauer G. Accurate Prediction of Biological Assays with High-Throughput Microscopy Images and Convolutional Networks. *J Chem Inf Model* 59, 1163-1171 (2019).

Reviewer #2:

Remarks to the Author:

The authors have successfully addressed all of my concerns. I have just one minor suggestion: It would be beneficial to include the following reference in your work - Wong et al. (Deep representation learning determines drug mechanism of action from cell painting images <https://doi.org/10.1039/D3DD00060E>).

REBUTTAL LETTER - MANUSCRIPT NCOMMS-23-14357A

Reviewer #1 (Remarks to the Author):

Summary: The authors present a study in which they predict HTS screening data based on high-content images.

General comments: The authors were only able to accommodate a few minor points. The main concern, which is lack of novelty, remains and my concerns have even aggravated.

Thank you for reviewing our revised manuscript and for your additional comments. We acknowledge that, despite extensive efforts, we have not been able to alleviate your concern regarding novelty, in our first revision. However, we respectfully disagree with the notion that only a few minor points were accommodated. Specifically, we incorporated all the minor points raised in the first review, including:

- *Addition of standard deviations and confidence intervals.*
- *Provision of additional insights into the predictive performance of image-based methods versus structure-based methods.*
- *Inclusion of the requested reference.*

Furthermore, we diligently addressed all major points from the initial review:

- *To address your concern on novelty, we included additional data and analyses, extensively revised significant sections of the manuscript, and, in accordance with your suggestions, explicitly clarified how we distinguish and build upon previous approaches. Despite our efforts, we acknowledge that this major point remains unresolved.*
- *We expanded and clarified the selection process of hyperparameters.*
- *We provided requested details on the chemistry-based method.*
- *To enhance relevance and value for the scientific community, we made our code accessible and incorporated results from applying our model to a publicly accessible dataset.*
- *We refined the technical terminology to improve the precision and clarity of our writing.*

We appreciate your thorough evaluation and have further refined our manuscript based on your feedback.

Major comments:

1) Lack of novelty: The presented study is still highly similar to reference [4] and [8]. In their rebuttal, the authors now point out the following four potential aspects of novelty:

a) Noisy single-point activity data sufficient (a few hundreds of points): This has been already shown in prior works (e.g. [4] and [8]) that this is sufficient. In current state-of-the-art methods, it is shown that it is possible for few-shot (less than 10 per activity class) or zero-shot [1] setting.

To our understanding, previous studies such as [4] and [8] have employed binary activity data derived from dose-response curves, expressed as $pXC50$ [4] or $IC50/EC50$ [8] of the given compound in the given assay. In these cases, each activity label is supported by multiple data points (usually 10-20), significantly enhancing the confidence in the labels. In addition, both previous approaches evaluated all assays at different thresholds of $pXC50/IC50/EC50$ (four in [4] and three in [8]), which allows for inclusion of more assays/data by identifying all splits that meet their criteria for number of active/inactive compounds. Consequently, a low level of noise in the labels is expected, with minimal occurrence of false-positive or false-negative labels.

REBUTTAL LETTER - MANUSCRIPT NCOMMS-23-14357A

In contrast, our approach employs noisy single-point activity data, where each binary activity label is based on a single data point from a single concentration of a compound in a single well of a bioassay, aligning with the prevalent practices of high-throughput screens. Consequently, our activity data is inherently noisier, incorporating a higher fraction of false-positive and false-negative labels.

Thus, we extend the work in [4] and [8] by showing that learning meaningful patterns from noisy, real-world, single-point activity data is possible, and results in similar predictive performance. Our findings indicate that, for many assays utilizing noisy single-point activity data, even when only a limited number of data points are available (resulting in a sparsely filled matrix), our model effectively learns and predicts the entire library's activity in that specific assay. This presents the potential to significantly reduce the number of compounds that need to undergo screening.

We have made revisions in the relevant sections to provide further clarification on this point. Kindly refer to lines 126 to 128 and lines 391 to 403.

Thank you for bringing reference [1] Sanchez-Fernandez et al. to our attention, was published only after our revision. We had cited a previous version from bioRxiv in our manuscript but now updated and cite this reference in our revised manuscript in lines 163, 170, and 941.

b) Imaging modality: the authors mention that they include bright field microscopy images. While this is an interesting observation, this type of transfer from pre-training on one modality (e.g. natural images) and then transferring to new modalities (e.g. biomedical images) is well-known and studied since the advent of deep convolutional networks ([2],[3]).

There appears to be a potential misunderstanding, and upon careful reflection, we acknowledge that, while clearly explained in the manuscript, the phrasing in our initial rebuttal letter may lead to confusion. The statement, 'We extend the scope of accurate predictions beyond fluorescent images to include brightfield images,' may not fully convey our intended meaning. To clarify, our approach does not merely involve the inclusion of brightfield images or their use for demonstrating transfer learning. Instead, we directly showcase the prediction of bioactivity from brightfield images. Our data suggests that a significant amount of phenotypic information regarding compound perturbations is present in brightfield images. Consequently, this implies that fluorescent images may not be a prerequisite, broadening the applicability of our methodology and potentially significantly reducing implementation costs.

For a detailed exploration of this aspect, please refer to our manuscript lines 217-233, where we report our findings. Additionally, we have extended the discussion lines 450-455 to further clarify this point.

c) Validation in in-vitro experiments: has also been done in [4].

We are aware of the experimental follow up conducted by Simm and colleagues, and it has not been our intention implying, that we provide the very first follow up of image-based bioactivity predictions in in vitro experiments. Our in vitro validation extends in scope, thereby enhancing its conclusions and value for the scientific community. Our work is not a critique of the work by Simm and colleagues, rather it builds on their and others' important findings.

The work by Simm and colleagues establishes the validity of bioactivity predictions, by in vitro testing predictions for (1) a kinase target in oncology, testing the 342 highest ranking compounds predictions; and (2) for a non-kinase target in CNS disease, testing 141 high-ranking compounds selected from clusters based on structure. Both assays had a very high predictive performance of

REBUTTAL LETTER - MANUSCRIPT NCOMMS-23-14357A

ROC-AUC >0.9. Simm and colleagues calculated enrichment by comparing the HTS hit rate with the hit rate in the follow-up assay.

To rigorously validate our predictions and understand the expected level of enrichment and its relation to the primary assay, we carefully selected four diverse assays. These assays cover different target classes and exhibit representative performances in the primary assay, ranging from a low ROC-AUC of 0.68 to a very high performance of 0.91. To ensure a reliable assessment of enrichment, we tested a minimum of 1000 compounds for each follow-up assay.

It's notable that in a real-world drug discovery setting, the conditions between a HTS assay and a follow-up, respectively DMTA assay (design-make-test-analyze), differ. For example, in HTS a small fraction of compounds may undergo degradation or alterations, resulting in false negatives. This is generally accepted since core compound structures are often represented by several close analogues in the library. Moreover, the quality control for specific compounds, typically provided by a compound management function, tends to be higher for follow-up assays than for HTS libraries. Even if essentially the same assay protocol is used for HTS and follow-up, subtle differences exist, making a follow-up assay generally more robust and reliable. For instance, HTS often involves a higher or different level of automation compared to follow-up assays. Follow-up assays may include different assay volumes, different reagent concentrations, different read-out format, additional controls and references, more data points per compound (replicates, dose-response), and more meticulous manual quality control of the data.

Therefore, we chose to sample a large fraction of compounds for validation i.e., more than 1,000 for each assay, and we employed a dual sampling approach, selecting a majority of compounds from the top-ranked 5% as well as choosing at least 500 compounds uniformly at random. This methodology enables the calculation of enrichment using bootstrapping. It is worth noting however, that sampling a large fraction of compounds imposes limits on the theoretical maximum achievable enrichment with the bootstrapping approach. For instance, by sampling the top 5% the maximum achievable enrichment rate is 20-fold. Conversely, sampling only the top 0.5% the maximum achievable enrichment rate would be 200-fold.

To further clarify our approach to in vitro validation we have provided additional key metrics in supplementary figure 10, expanded on the methods used for follow-up screening and calculation of enrichment (lines 677-698) and further discuss the context and significance of our work regarding that point in lines 433-447.

d) Comprehensive analysis: this is indeed slightly novel, but not a radically new finding. QSAR models are often analyzed in this way. The authors should focus on a single, clear and relevant novel aspect of their work and write the paper with this novelty as main theme of the paper.

We appreciate your recognition of the value in our analysis, and we believe it holds significance for many in the scientific community. While recognizing that other QSAR models have undergone similar analyses, our distinctive contribution lies in presenting a first, comprehensive analysis specifically for Cell Painting-based bioactivity prediction. In doing so, we address fundamental questions surrounding its applicability, such as the robustness of Cell Painting-based QSAR models across diverse assay types, technologies, and target classes. And we explore whether specific assay and target characteristics influence the suitability of Cell Painting-based bioactivity prediction. At a general level, our analysis expands our understanding of the strengths and weaknesses of Cell Painting-based bioactivity prediction. More tangibly, this information offers valuable guidance for the implementation of Cell Painting-based bioactivity prediction in early drug discovery. Moreover, we believe that these insights can steer further refinements in the models,

REBUTTAL LETTER - MANUSCRIPT NCOMMS-23-14357A

contributing to ongoing advancements in the field.

Based on your feedback in comments 1a) to d) we have again revised several sections of the manuscript to further underscore the novelty and impact of our work e.g., at lines 126-128, lines 391-403, lines 433-447, lines 450-455, and addition of supplementary figure 10 at lines 960-961.

2) The study is smaller than previous studies with only 140 assays analyzed while [4] already modelled ~1200. The authors should benchmark their method on previously established datasets and benchmarks, concretely the benchmark proposed in [8] and [1]. The authors should use datasets within a size of state-of-the-art methods.

In accordance with your recommendations, we conducted a benchmark of our method using the datasets established and utilized by both Hofmarcher et al. and Sanchez-Fernandez et al. Our model demonstrated an average performance of ROC-AUC of 0.732 ± 0.222 . This aligns closely with the reported performances of the supervised ResNet model by Hofmarcher et al. (0.731 ± 0.19 ROC-AUC) and the linear probing contrastive learning model 'CLOOME,' as recently documented on the same dataset (0.714 ± 0.20 ROC-AUC). The detailed results of these additional experiments are provided in Supplementary Figure 8, lines 941-951, and lines 159-170.

3) Reproducibility: has improved, but is still low because the pre-processed dataset has not been provided. The authors should provide the pre-processed dataset in simple downloadable and accessible form, such that other researchers can easily reproduce their findings.

Thank you for bringing up this concern. We recognize that we may have underestimated the complexity involved in downloading the dataset. To address this, we have now introduced an additional script along with a comprehensive step-by-step guide for the automated download and pre-processing of the "ChEMBL dataset", available at <https://github.com/cfredinh/bioactive>

This resource aims to streamline the process and enhance accessibility for the scientific community.

References:

- [1] Sanchez-Fernandez, A., Rumetshofer, E., Hochreiter, S., & Klambauer, G. (2023). CLOOME: contrastive learning unlocks bioimaging databases for queries with chemical structures. *Nature Communications*, 14(1), 7339.
- [2] Yu, Y., Lin, H., Meng, J., Wei, X., Guo, H., & Zhao, Z. (2017). Deep transfer learning for modality classification of medical images. *Information*, 8(3), 91.
- [3] Kim, H. E., Cosa-Linan, A., Santhanam, N., Jannesari, M., Maros, M. E., & Ganslandt, T. (2022). Transfer learning for medical image classification: a literature review. *BMC medical imaging*, 22(1), 69.
- [4] Simm J, et al. Repurposing High-Throughput Image Assays Enables Biological Activity Prediction for Drug Discovery. *Cell Chem Biol* 25, 611-618 e613 (2018).
- [8] Hofmarcher M, Rumetshofer E, Clevert DA, Hochreiter S, Klambauer G. Accurate Prediction of Biological Assays with High-Throughput Microscopy Images and Convolutional Networks. *J Chem Inf Model* 59, 1163-1171 (2019)

REBUTTAL LETTER - MANUSCRIPT NCOMMS-23-14357A

Reviewer #2 (Remarks to the Author):

The authors have successfully addressed all of my concerns. I have just one minor suggestion: It would be beneficial to include the following reference in your work - Wong et al. (Deep representation learning determines drug mechanism of action from cell painting images <https://doi.org/10.1039/D3DD00060E>).

We are delighted to have successfully addressed your concerns and express our sincere gratitude for acknowledging and appreciating our efforts. We firmly believe that, with your feedback, along with that of the other reviewer and the editor, our manuscript has significantly strengthened in terms of clarity and scientific novelty.

Your mention of Wong et al.'s recent work is duly noted. We have incorporated this citation in our revised manuscript at line 427.

Thank you once again for your invaluable contributions to enhancing the quality of our research.